# Regulation and Role of Neuron-Derived Hemoglobin in the Mouse Hippocampus

**DOI:** 10.3390/ijms23105360

**Published:** 2022-05-11

**Authors:** Yujiao Lu, Jing Wang, Fulei Tang, Uday P. Pratap, Gangadhara R. Sareddy, Krishnan M. Dhandapani, Ana Capuano, Zoe Arvanitakis, Ratna K. Vadlamudi, Darrell W. Brann

**Affiliations:** 1Department of Neurosurgery, Medical College of Georgia, Augusta University, Augusta, GA 30912, USA; yulu@augusta.edu (Y.L.); kdhandapani@augusta.edu (K.M.D.); 2Department of Neuroscience and Regenerative Medicine, Medical College of Georgia, Augusta University, Augusta, GA 30912, USA; jinwang@augusta.edu (J.W.); ftang@augusta.edu (F.T.); 3Department of Obstetrics and Gynecology, University of Texas Health, San Antonio, TX 78229, USA; pratap@uthscsa.edu (U.P.P.); sareddy@uthscsa.edu (G.R.S.); vadlamudi@uthscsa.edu (R.K.V.); 4Rush Alzheimer’s Disease Center, Rush University Medical Center, Chicago, IL 60612, USA; ana_capuano@rush.edu (A.C.); zoe_arvanitakis@rush.edu (Z.A.)

**Keywords:** hemoglobin, hypoxia, brain, cerebral ischemia, aging, neuron

## Abstract

Hemoglobin (Hb) is the oxygen transport protein in erythrocytes. In blood, Hb is a tetramer consisting of two Hb-alpha (Hb-α) chains and two Hb-beta (Hb-β) chains. A number of studies have also shown that Hb-α is also expressed in neurons in both the rodent and human brain. In the current study, we examined for age-related regulation of neuronal Hb-α and hypoxia in the hippocampus and cerebral cortex of intact male and female mice. In addition, to confirm the role and functions of neuronal Hb-α, we also utilized lentivirus CRISPR interference-based Hb-α knockdown (Hb-α CRISPRi KD) in the non-ischemic and ischemic mouse hippocampus and examined the effect on neuronal oxygenation, as well as induction of hypoxia-inducible factor-1α (HIF-1α) and its downstream pro-apoptotic factors, PUMA and NOXA, and on neuronal survival and neurodegeneration. The results of the study revealed an age-related decrease in neuronal Hb-α levels and correlated increase in hypoxia in the hippocampus and cortex of intact male and female mice. Sex differences were observed with males having higher neuronal Hb-α levels than females in all brain regions at all ages. In vivo Hb-α CRISPRi KD in the mouse hippocampus resulted in increased hypoxia and elevated levels of HIF-1α, PUMA and NOXA in the non-ischemic and ischemic mouse hippocampus, effects that were correlated with a significant decrease in neuronal survival and increased neurodegeneration. As a whole, these findings indicate that neuronal Hb-α decreases with age in mice and has an important role in regulating neuronal oxygenation and neuroprotection.

## 1. Introduction

Hemoglobin (Hb) is an iron-containing metalloprotein located in red blood cells whose primary function is to carry oxygen (O_2_) from the lungs to various tissues of the body [1]. Hb is a tetramer of two α-globin and two β-globin chains, with each monomeric chain containing a heme moiety for O_2_ transport. The affinity of Hb monomers for O_2_ is higher than that of α2β2 tetramers, and thus assembly is not required for Hb monomer function [2,3,4]. Recently, Hb monomers have been shown to be expressed in many tissues other than blood, including alveolar epithelial cells [5], endometrial cells [6], macrophages [7], retinal pigment cells [8], and in neurons in several brain regions, including the hippocampus, cerebral cortex and substantia nigra [9,10,11]. Hb overexpression in cultured dopamine neurons was shown to modulate the expression of genes that regulate O_2_ homeostasis and oxidative phosphorylation [11]. Furthermore, systemic injection of erythropoietin, which is induced by hypoxia, was shown to enhance Hb-α mRNA and reduce tissue hypoxia in the hypoxic mouse brain [9], suggesting that neuronal Hb-α may regulate neuronal oxygenation. Neuronal Hb-α is also significantly decreased in neurodegenerative disorders such as Alzheimer’s disease, Parkinson’s disease and dementia with Lewy bodies [12]. Since the brain has one of the highest demands for O_2_, it is suggested that neuronal Hb-α may serve as a O_2_ transport/storage factor in neurons so as to help buffer deficits and peak elevations of O_2_ consumption and may help protect neurons from stressors and neurodegeneration [13].

To further enhance understanding of the regulation and role of neuronal Hb-α in the forebrain, we examined whether there are age-related changes in Hb-α expression in the intact male and female mouse hippocampus and determined whether the observed changes in neuronal Hb-α expression correlate with forebrain hypoxia. We examined the hippocampal CA1 and CA3 regions, as the hippocampus is well known to be highly susceptible to ischemic damage under hypoperfusion [14]. In addition, we established lentivirus CRISPR interference-based Hb-α knockdown (Hb-α CRISPRi KD) and examined the effect of Hb-α knockdown on neuronal oxygenation in the mouse hippocampus basally and after cerebral ischemia. Finally, we also examined whether Hb-α CRISPRi KD altered expression of the hypoxia-inducible transcription factor-1α (HIF-1α) and its downstream pro-apoptotic factors, p53-upregulated modulator of apoptosis (PUMA) and NOXA (Latin for damage) [15,16] in the hippocampus basally and after ischemia, and examined whether neuronal Hb-α CRISPRi KD affected hippocampal neuronal survival and/or neurodegeneration. The results of the study demonstrate that there is an age-related decrease in neuronal Hb-α expression in the hippocampus and cerebral cortex that correlates with an increase in hypoxia in these brain regions. Furthermore, the results of our study provide evidence of an important role for neuronal Hb-α in regulating hippocampal neuronal oxygenation in non-ischemic and ischemic conditions and indicate that neuronal Hb-α is neuroprotective in the hippocampus.

## 2. Results

To determine whether there are age-related changes in neuronal Hb-α expression and hypoxia in the forebrain, we collected hippocampal and cortical tissue sections from 6-, 12- and 18-month-old intact male and female mice and performed immunohistochemistry using an antibody to Hb-α. To measure forebrain hypoxia, mice received an intra-peritoneal injection of the hypoxia marker, pimonidazole (hypoxyprobe-1) (60 mg/kg) 1 h before sacrifice, which detects tissue hypoxia (pO_2_ < 10 mm Hg) using an immunohistochemical technique [9,17,18]. Pimonidazole is reductively activated in hypoxic cells, and the activated intermediate forms stable covalent adducts with thiol (sulfhydryl) groups in proteins, peptides and amino acids, which can be detected using a monoclonal antibody that binds these adducts [19]. As shown in Figure 1 and Figure 2, representative photomicrographs (**Panel A**) and semi-quantitative intensity analysis from all mice (**Panel B**) are presented for Hb-α staining and pimonidazole staining in the hippocampal CA1 and CA3 region and cerebral cortex of intact male mice (Figure 1) and intact female mice (Figure 2) at 6, 12 and 18 months of age. Semi-quantitative intensity analysis of neuronal Hb-α immunostaining in the hippocampal CA1 and CA3 regions and cerebral cortex of all intact male mice revealed that neuronal Hb-α levels are high at 6 months age, followed by a modest decrease at 12 months and a robust decrease at 18 months of age (Figure 1B). A similar age-related decrease in Hb-α immunoreactive levels was also observed in the hippocampal CA1 and CA3 regions and cerebral cortex of intact female mice (Figure 2B). In contrast, semi-quantitative intensity analysis of pimonidazole staining showed an inverse relationship to neuronal Hb-α expression, with hypoxia initially low in the hippocampal CA1, CA3 and cortex at 6 months of age, followed by a significant increase at 12 and 18 months of age in both intact male mice (Figure 1B) and intact female mice (Figure 2B). Correlation coefficient analysis between neuronal Hb-α and pimonidazole in the hippocampal CA1, CA3 and cortex of male (Figure 1C) and female (Figure 2C) mice was further conducted, which revealed a strong negative correlation between neuronal-derived Hb-α and brain hypoxia. These results demonstrate that an age-related fall in neuronal Hb-α expression is correlated with an increase in forebrain hypoxia in both male and female mice. We also examined for gender differences in neuronal Hb-α levels and hypoxia in the hippocampus and cortex during aging. As shown in Figure 3A–C, semi-quantitative intensity analysis revealed that males had higher neuronal Hb-α levels in the hippocampal CA1 and CA3 regions and cerebral cortex at all ages as compared to female mice. Males also had generally lower brain hypoxia than females in the three regions, but this was only significant in the hippocampal CA1 region (Figure 3D–F).

To help determine the roles and functions of neuronal Hb-α in the hippocampus, we established in vivo lentivirus CRISPR interference-based Hb-α knockdown (Hb-α CRISPRi KD). The CRISPR-Cas9 system has been widely used as a powerful tool for genome editing in cells and animal tissue, including post-mitotic brain neurons [20]. In the CRISPRi approach modified from the CRISPR-Cas9 system, an engineered nuclease null form Cas9 (dCas9) is fused to the transcription repressor KRAB (dCas9-KRAB). Navigated by a single guide RNA (sgRNA), dCas9-KRAB can recognize specific genomic loci of interest and block transcription around a window of the transcriptional start site (TSS), thereby providing effective knockdown of the target gene [21]. To establish the feasibility of this approach, we performed preliminary studies to use the plasmid constructs carrying the dCas9-KRAB modified from the lentiCRISPR v2 vector (Figure 4A). Three sgRNAs were designed to target the DNA region from −50 to 300 bp relative to the TSS of the Hb-α encoding gene HBA1/2. The effectiveness of the Hb-α CRISPRi KD was then examined in mouse HT-22 cells and cultured primary mouse hippocampal neurons. As shown in Figure 4, Hb-α CRISPRi KD was demonstrated to strongly inhibit the expression of Hb-α in mouse HT-22 cells (Figure 4B) and primary cultured mouse hippocampal neurons transfected with the sgRNA-dCas9-KRAB plasmid as indicated by GFP expression (Figure 4C).

Next, we further generated lentiviral particles carrying the scramble or sgRNAs of HBA1/2 (Scramble/HBA1/2 lenti-CRISPRi) and validated the efficiency of the CRISPRi system to knockdown Hb-α expression in vivo by stereotactic lentivirus infusion in the male mouse hippocampus (Figure 5). Figure 5A provides a schematic illustration of the conditional CRISPRi system used to selectively suppress Hb-α expression in the hippocampal CA1. Figure 5B contains representative IHC results for Hb-α in the hippocampal CA1 region at 2 weeks after infusion with scramble (upper) or HBA1/2 (lower) lenti-CRISPRi. The IHC results revealed that the HBA1/2 lenti-CRISPRi strongly downregulated Hb-α expression in the hippocampal CA1 and CA3 regions at 2 weeks post lentivirus infusion (Figure 5B–F). Western blot analysis also confirmed that hippocampal Hb-α levels were strongly decreased in the HBA1/2-targeted lenti-CRISPRi infusion groups, as compared with the scramble group and non-infused contralateral groups at 2 weeks after infusion (Figure 5C). Finally, as shown in Figure 5D–G, IHC for Hb-α and pimonidazole staining confirmed the effective knockdown of Hb-α expression, which resulted in increased hypoxia in the hippocampal CA1 and CA3 regions. Note that the scramble lenti-CRISPRi had no effect on Hb-α expression or hypoxia. These results indicate that neuronal Hb-α has a role in regulating neuronal oxygenation in the hippocampus under basal conditions.

HIF-1α is a major transcription factor that is induced by hypoxia [22,23]. HIF-1α-induced genes such as PUMA and NOXA have been implicated in mediating neuronal cell death after ischemia or injury [16,24,25]. Therefore, we next examined whether knockdown of Hb-α in the hippocampus of the 3-month-old male mouse leads to increased HIF-1α, PUMA and NOXA after GCI. We employed three groups in the study: (1) a “control” group that received no injection, (2) a “CRISPRi-scramble” group that received lenti-CRISPRi-scramble injection in the hippocampus, and (3) a “CRISPRi-HBA1/2” group that received lenti-CRISPRi-HBA1/2 injection in the hippocampus to knockdown Hb-α. We also used a sham control group that did not receive GCI. Animals were examined at 7 days after GCI and IHC were performed on hippocampal sections for HIF-1α, PUMA and NOXA. As shown in Figure 6A,B, the results revealed that knockdown of Hb-α in the hippocampus led to a small but significant increase in HIF-1α expression in sham animals in both the hippocampal CA1 and CA3 regions, while the CRISPRi-scramble control had no effect. Notably, lenti-CRISPRi knockdown of Hb-α led to a robust increase in HIF-1α expression in the CA1 and CA3 regions at 7 days after GCI, which was highly significant as compared to the GCI control and GCI + CRISPRi-scramble control (*p* < 0.01). Examination of the HIF-1α regulated genes, PUMA and NOXA, revealed that knockdown of Hb-α in the hippocampus led to a small but significant increase in PUMA and NOXA expression in sham animals in both the hippocampal CA1 and CA3 regions, while the CRISPRi-scramble control had no effect (Figure 7A–D and Figure 8A–D). Notably, similar to the HIF-1α results, lenti-CRISPRi knockdown of Hb-α led to a robust increase in PUMA and NOXA expression in the hippocampal CA1 and CA3 regions at 7 days after GCI, which was highly significant as compared to GCI control and GCI + CRISPRi-scramble control (*p* < 0.01) (Figure 7A–D and Figure 8A–D).

Since PUMA and NOXA are pro-apoptotic genes, their enhanced induction in neuronal Hb-α knockdown mice could suggest that Hb-α is neuroprotective, and its loss leads to increased neuronal cell damage and death basally and after ischemia. Therefore, we next examined the effect of neuronal Hb-α knockdown on neuronal survival and neurodegeneration in the hippocampus. To accomplish this, we used IHC for the neuron marker, NeuN, to measure neuron survival, and we used Fluoro Jade C (FJC) staining, a marker of neurodegeneration, to measure neurodegeneration in sham and GCI animals at 7 days post GCI (GCI-7d). Representative photomicrographs are shown in Figure 9A and Figure 10A, while NeuN-positive cell number and FJC intensity analysis in the hippocampal CA1 region for all animals are shown in Figure 9B,C and Figure 10B,C. The results revealed that Hb-α knockdown led to a mild but significant decrease in NeuN staining and NeuN-positive cell number, and a mild but significant increase in FJC staining in sham animals in both the hippocampal CA1 and CA3 regions, while the CRISPRi-scramble control had no effect (Figure 9 and Figure 10A–C). These results indicate that there is a mild decrease in neuronal survival and a mild increase in neurodegeneration after Hb-α knockdown in non-ischemic animals. Furthermore, similar to the PUMA and NOXA results, lenti-CRISPRi knockdown of Hb-α led to a robust decrease in neuronal survival (as evidenced by decreased NeuN staining and NeuN-positive cell number) and a robust increase in neurodegeneration (as evidenced by increased FJC staining) in the hippocampal CA1 and CA3 regions at 7 days after GCI, which was highly significant as compared to the GCI control and GCI + CRISPRi-scramble control (*p* < 0.01) (Figure 9 and Figure 10A–C). This finding suggests that neuronal Hb-α plays an important role in protecting neurons from ischemic/hypoxia-induced cell damage and death.

## 3. Discussion

The current study revealed several novel findings. First, it demonstrates that there is an age-related decrease in neuronal Hb-α immunoreactive levels in the hippocampal CA1 and CA3 regions and cerebral cortex in both intact male and female mice, which is correlated with an age-related increase in hypoxia in these brain regions. Second, it provides evidence of a role of neuronal Hb-α in regulating neuronal oxygenation, as lenti-CRISPRi knockdown of neuronal Hb-α resulted in increased hypoxia in the hippocampal CA1 and CA3 regions of young mice. Third, lenti-CRISPRi knockdown of neuronal Hb-α led to increased expression of the transcription factor, HIF-1α, and its downstream pro-apoptotic target genes, PUMA and NOXA, in the hippocampus both basally and after GCI. Fourth, a neuroprotective role for neuronal Hb-α was suggested by the finding that in vivo lenti-CRISPRi knockdown of neuronal Hb-α resulted in enhanced neurodegeneration and decreased neuronal survival in the hippocampus both basally and after cerebral ischemia.

In this study, we used IHC to detect neuronal Hb-α in the perfused mouse brain and in cultured neurons. We found significant Hb-α expression in neurons in the mouse hippocampal CA1 and CA3 regions, cerebral cortex, and in cultured hippocampal neurons. This finding agrees with previous work using in situ hybridization, RT-PCR, Western blot analysis and IHC, which demonstrated strong expression of Hb-α in neurons in the hippocampus and cerebral cortex of the mouse, rat and human [9,10], and in cultured neurons [9]. Confirmation of the specificity of the Hb-α antibody used in our study was achieved by lenti-CRISPRi knockdown of neuronal Hb-α, which essentially abolished Hb-α immunostaining in transfected neurons in vitro and in vivo. Immunoblot of hippocampal tissue samples also confirmed loss of Hb-α in knockdown samples. A sex difference was noted in neuronal Hb-α levels, as female mice had lower levels than males in the hippocampal CA1 and CA3 regions and cerebral cortex at all ages examined. The reason for the gender difference is unclear, but it could be due to differences in sex steroids, sex chromosomes, or erythropoietin, a major regulator of Hb. Along these lines, a previous work demonstrated that females have 12% lower Hb in venous blood than males, which is thought to be due to sex steroids [26]. However, a search of the literature failed to reveal comparable studies on whether sex steroids regulate neuronal Hb expression. Previous studies showed that injection or transgenic overexpression of erythropoietin in mice strongly upregulates neuronal Hb-α expression in the brain [9]. Furthermore, the erythropoietin upregulation of neuronal Hb-α expression was correlated with enhanced brain oxygenation under both physiological and hypoxic conditions, which further supports the findings of our study using lenti-CRISPRi knockdown of neuronal Hb-α. While erythropoietin is classically produced in the kidney, its production also occurs in extrarenal tissues, including the brain [27]. Sex differences in extrarenal production of erythropoietin have been reported previously in rodents, with adult males having higher extrarenal production than adult females [28]. Thus, differences in extrarenal production of erythropoietin might underlie the sex differences we observed in neuronal Hb-α expression. Further studies are needed to address this possibility.

An additional finding of our study was that the aging mouse hippocampus and cortex exhibited mild hypoxia, which was most pronounced at 18-months of age in both male and female mice. Mild hypoxia appeared higher in females in all three brain regions at all ages, but this was only statistically significant in the hippocampal CA1 region. In agreement with our results, Snyder et al. [29], using samples from both male and female mice, also reported that the aged mouse hippocampus displays signs of chronic mild hypoxia at 18 and 24 months of age, as evidenced by a significant elevation in HIF-1α expression. Mild hypoxia in the aged mouse brain could be due to either the decreased neuronal Hb-α levels observed in our study or to a potential age-related decrease in brain perfusion. However, several studies have shown that there is no significant change in cerebral blood flow during aging in mice [30,31]. Thus, the decreased neuronal Hb-α expression observed in our study could potentially have a role in the increased forebrain hypoxia in aged mice. In support of this possibility, we found that lenti-CRISPRi knockdown of neuronal Hb-α in the hippocampus similarly resulted in mild hypoxia in the hippocampus of young mice. The age-related changes in neuronal Hb-α and hypoxia in the hippocampus and cortex could contribute to the well-known increased sensitivity of these regions to ischemia and trauma in aged animals [32,33,34].

Functionally, acute hypoxia has been shown to cause depression in synaptic activity in the brain, while more prolonged hypoxia can lead to neuronal cell loss and damage [35]. Two weeks after lenti-CRISPRi knockdown of neuronal Hb-α, we observed a mild decrease in neuronal survival and an increase in neuronal damage in the non-ischemic mouse hippocampus basally. In contrast, lenti-CRISPRi knockdown of neuronal Hb-α in the GCI model led to a robust enhancement of neuronal loss and neuronal damage in the ischemic hippocampus as compared to GCI control animals. This effect appeared specific, as no effect on neuronal survival or neurodegeneration was observed in GCI + CRISPRi-scramble control mice. Furthermore, we also observed increased expression of the hypoxia-inducible transcription factor, HIF-1α, and its downstream proapoptotic target genes, PUMA and NOXA, both basally and after ischemia in lenti-CRISPRi Hb-α knockdown animals. Previous work by our group and others demonstrated that delayed cell death in the hippocampus following GCI involves apoptosis and that HIF-1α, PUMA and NOXA are all elevated in the hippocampal CA1 region following GCI [36,37,38,39,40]. Based on these findings, we propose that the enhanced elevation of HIF-1α, PUMA and NOXA in the hippocampus of lenti-CRISPRi neuronal Hb-α knockdown mice could contribute to the increased neuronal loss and damage we observed in the knockdown mice. In support of this suggestion, Helton et al. [41] demonstrated a pro-apoptotic role for HIF-1α in the hippocampus after GCI, as brain-specific HIF-1α knockout mice exhibited significant downregulation of pro-apoptotic genes in the hippocampus and were protected from GCI. Furthermore, both PUMA and NOXA have been implicated in mediating hypoxia- and HIF-1α-induced cell death in neurons. For instance, siRNA knockdown of PUMA prevented HIF-1α-induced cell death in hippocampal neuronal cells [15], while knockdown of NOXA rescued cells from hypoxia-induced cell death and decreased infarct volume in stroke [16]. Mechanistically, the decreased neuronal oxygenation following loss of neuronal Hb-α could make hippocampal neurons more sensitive to damage from various stressors and thus result in greater neuronal loss and damage following insults such as ischemia. Furthermore, since Hb is also a nitric oxide scavenger [42,43], loss of neuronal Hb could potentially lead to greater oxidative stress and nitrosative stress in the hippocampus, which could also contribute to the increased neuronal loss and damage we observed in Hb-α knockdown mice. Further studies are needed to address this possibility.

In conclusion, the current study provides evidence of age-related and sex-dependent changes in neuronal Hb-α expression in the forebrain. It also demonstrates an important role for neuronal Hb-α in regulating neuronal oxygenation and neuronal survival in the hippocampus, both basally and after ischemic insult. As a whole, the findings add to a growing body of evidence that Hb expression in non-erythroid cells has important functions in a variety of tissues, including the brain.

## 4. Materials and Methods

### Animals

All animal experiments were approved by the Augusta University Institutional Animal Care and Use Committee (Protocol #2019-0998) and were performed in accordance with the National Institutes of Health Guide for the Care and Use of Laboratory Animals. Male and female C57BL/6J mice at 6 months, 12 months and 18 months (Jackson Laboratory, Stock No. 000664) were purchased from Jackson Laboratory for the aging related studies. Three-month-old male mice were also purchased from Jackson Laboratory for the Hb-α knockdown studies. Mice were group-housed (5 mice maximum cage capacity) in environmentally controlled conditions with 12:12/h light/dark cycle and were provided with food and water ad libitum.

## 5. Plasmid Construction and sgRNA Design

The lentiviral vector used for this study was constructed from pLV hU6-sgRNA hUbC-dCas9-KRAB-T2a-GFP (Addgene, Watertown, MA, USA, #71237). Plasmid pLJM1-EGFP was used as a control. sgRNAs were designed with an online web tool: https://zlab.bio/guide-design-resources (Previous: http://crispr.mit.edu/, accessed on 3 July 2018) [44]. Three sgRNAs were designed to target the DNA region from −50 to 300 bp relative to the transcription start site of the Hb-α encoding gene HBA1/2: sgRNA #1: 5′-gcaacatcaaggctgcctgg-3′; sgRNA #2: 5′-gtccagagaggcatgcaccg-3′; and sgRNA #3: 5′ atggccaccaatcttccccc-3′.

## 6. Lentivirus Production

HEK293FT cells were maintained in Dulbecco’s Modified Eagle Medium (DMEM, Gibco, Waltham, MA, USA, Cat. 11965092) in 10% FBS (Gibco, Waltham, MA, USA, Cat. 26140), 1% P/S (penicillin/streptomycin in 10,000 units solution, Gibco, Waltham, MA, USA, Cat#15140122) with 2 mM Glutamax (Gibco, Waltham, MA, USA, Cat# 35050061). lentiviral particles were generated by co-transfecting HEK293FT cells with virus packing vectors psPAX2 (Addgene, Watertown, MA, USA, Cat#12260) and pVSVG (Addgene, Watertown, MA, USA, Cat#12259), with lenti-CRISPRi plasmid. PEImax (Polyscience, Philadelphia, PA, USA, Cat#24765-1) was used as a transfection reagent. The medium was changed 8 h after transfection. The virus supernatant was harvested at 48 and 60 h and kept at 4 °C until used. The collected supernatant was centrifuged for 5 min at 1000× *g* at 4 °C to pellet the debris and was passed through a 0.45 polyethesulfone (PES) filter before it was concentrated. The media were ultracentrifuged at 25,000 rpm (52,616–69,875 g) using a Beckman ultracentrifuge with a SW28 rotor. The virus pellet was resuspended with 1× phosphate buffered saline (PBS). The pooled particles were loaded into 1.5 mL 20% sucrose in SW55 tubes and centrifuged at 24,500 rpm (40,802–72,812 g) at 4 °C for 90 min in a Beckman ultracentrifuge. The virus was then stored in a final volume of 100 μL in PBS. The titers of concentrated lentivirus in our experiment could reach 1 × 10^10^ IU/mL.

## 7. Cell Culture and Transfection

HT22 cells and primary neurons were cultured for testing the efficacy of Hb-α CRISPRi to inhibit the expression of Hb-α. The HT22 cell line was obtained from Millipore Sigma (St. Louis, MO, USA, SCC129). The HT22 cells were expanded in high glucose DMEM with 10% FBS, and 1×L-Glutamine. Cells were infected with lentivirus on day 2 after splitting. Mouse hippocampal neuron cultures were performed as previously described [45]. Briefly, the hippocampi from C57BL/6 mice on embryonic day 17.5–18.5 (E 17.5–18.5) were micro-dissected, dissociated, and seeded at a density of ~2 × 10^5^/mL in poly-D-lysine (Sigma-Aldrich, St. Louis, MO, USA, Cat# P7280) coated plates, followed by incubation at 37 °C in a humidified atmosphere of 5% CO_2_ in the air. Primary neurons were infected with lentivirus at 3 days in vitro (DIV) and were subjected to evaluation at 10–14 DIV.

## 8. Lentivirus Infusion

The titer of the lentivirus used for intracerebral stereotactic infusion was 1 × 10^10^ infectious Unit (IU) per ml. Following anesthesia with a mixture of O_2_ and isoflurane, C57BL/6 mice were stereotactically infused with lentivirus-encoding Hb-α CRISPRi-dCas9-KRAB or scramble. The lentivirus was infused bilaterally into the hippocampal region. Virus was microinjected into two sites in the CA1 and CA3 regions (2 µL per site) on the left and right hippocampus (site 1: anterior–posterior (AP)—2.0 mm, medio-lateral (ML) ± 1.5 mm, dorsoventral (DV)— 1.5 mm from the bregma; site 2: AP—2.06 mm, ML ± 2.35 mm, DV—2.35 mm relative to bregma). The viral solution in a glass cannula was infused using a microinjection pump (Harvard Apparatus, Holliston, MA, USA) at a flow rate of 0.1 μL/min into each hemisphere sequentially. Animals were sacrificed after 2 weeks, and the brain tissue was collected for evaluation.

## 9. Hypoxia

The hypoxia level of the brain tissue was detected with a Hypoxyprobe Kit (pimonidazole based) from Hypoxyprobe, Inc. (Cat# HP1-100 Kit, Hyproxyprobe, MA, USA) following the manufacturer’s instruction. Briefly, the male or female C57BL/6 mice received an intra-peritoneal injection of pimonidazole (60 mg/kg) 60 min before tissue collection. Hypoxia level was measured by staining the tissue sections with anti-pimonidazole mouse IgG monoclonal antibody and immunohistochemistry analysis.

## 10. Global Cerebral Ischemia

Two-vessel occlusion global cerebral ischemia (GCI) was performed on WT mice as described previously [46]. Then, 2 mg of carprofen (Rimadyl, Bio-Serv, Flemington, NJ, USA, formulated in a 5 g food tablet) was given to each mouse 8 h before surgery and after surgery for analgesia. Mice were anesthetized with isoflurane (4% induction, 1–2% maintenance). Via a midline incision of the neck, common carotid arteries (CCAs) on both sides were exposed and encircled loosely with 4-0 sutures to enable later occlusion with an aneurysm clip. Global ischemia was induced by bilateral occlusion of the CCAs for 30 min. After ischemia, the clips were removed, and reperfusion was confirmed by visual inspection of blood flow in CCAs. Five minutes after reperfusion, the incisions were closed with wound clips, and isoflurane was discontinued. The core body temperature of mice before, during, and after ischemia was maintained at 37 °C by a homeothermic blanket, with rectal temperature recorded continuously. Before being returned to their cage at room temperature, the mice were kept in a warm chamber at 33 °C for 2 h to maintain a body temperature of approximately 37 °C. For the sham controls, identical procedures were performed, and CCAs were exposed without occlusion. Mice that showed dilated pupils during GCI surgery and lost the righting reflex for at least 1 min after incision closure were included for further experiments.

## 11. Brain Section Collection and Immunohistochemistry (IHC) Analysis

Mice were sacrificed by an anesthetic overdose (5% isoflurane inhaled for 10 min) and transcardially perfused with ice-cold saline, followed by 4% paraformaldehyde before decapitation. The brains were removed, fixed overnight in 4% paraformaldehyde, and cryoprotected with 30% (*w*/*v*) sucrose in PBS for sectioning on a cryostat. Coronal sections (30 μm thick, for a total of ~60 sections) containing the hippocampal structure were collected between −0.94 and −2.80 mm from bregma. IHC was performed as previously described [45]. For each immunostaining, at least 4 randomly selected sections were chosen from each animal and processed for histological analysis. The following primary antibodies were used: NeuN (rabbit, 1:1000, Millipore, St. Louis, MO, USA, Cat# ABN78); GFP (chicken, 1:1000, Invitrogen, Waltham, MA, USA, Cat# A10262); Hemoglobin-α (rabbit, 1:500, Proteintech, Rosemont, IL, USA, Cat# 14537-1-AP); PUMA (rabbit, 1:800, Abcam, Cambridge, UK, Cat# ab9643); NOXA (mouse, 1:800, Abcam, Cambridge, UK, Cat# ab13654); Fluoro-Jade C (F-Jade C) (EMD Millipore, St. Louis, MO, USA, Cat# AG325-30MG). For IHC analysis, images were captured using a Zeiss 510 confocal microscope at 20× or 40× (Plan-Apochromat objective) for sections, and 63× for cells, with constant exposure for each marker in all analyzed sections. The hippocampal CA1 and CA3 from at least 3 different sections of each animal were used for analysis. Each data on the bar graph represents an average of each animal. The mean IF intensity of each marker was quantified with ImageJ software.

## 12. Tissue Lysates Preparation and Western Blot Analysis

Mice were sacrificed under deep anesthesia and were perfused with ice-cold 0.9% saline, and the hippocampus was isolated. Spleen tissue of the mice was obtained as a control sample for detecting Hb-α. The tissue samples were frozen in liquid nitrogen or kept on ice for immediate homogenization in RIPA buffer (50 mM Tris•HCl at pH 7.4, 150 mM NaCl, 1% Triton x-100, 1% sodium deoxycholate, 0.1%SDS, 1 mM EDTA) with complete protease inhibitor (Roche, Indianapolis, IN, USA, REF# 5892970001) and PhosphoSTOP (Roche, Indianapolis, IN, USA, REF# 04906845001) with a tissue tearor followed by denaturation at 95 °C for 10 min. Immunoblotting analysis was performed as previously described [45]. Primary antibodies used included Hemoglobin-α (rabbit, 1:5000, Proteintech, Rosemont, IL, USA, Cat# 14537-1-AP), and GAPDH (mouse, 1:2000, Santa Cruz Biotechnology, Dallas, TX, USA, Cat# sc-32233). Blots were visualized using a LI-COR Odyssey imager, and the ImageJ analysis software (Version 1.49; NIH, Bethesda, MD, USA) was used to determine the intensity of each band. Band densities for the indicated proteins were normalized to the corresponding loading controls.

## 13. Statistical Analysis

GraphPad Prism 6 software (GraphPad Software Inc., San Diego, CA, USA) was used to analyze all the data. Data are presented as mean ± SEM. For each statistical analysis, a normality test was performed using the Shapiro–Wilk test in Prism, and the data were normally distributed. T test was used to compare two groups. ANOVA tests were conducted to compare three or more groups. When the ANOVA test was found to be significant, the post hoc Dunnett’s test, Sidak’s multiple comparison test and Tukey’s test were conducted to make pairwise comparisons to determine the significance between the two groups. Correlation coefficient analysis between neuronal Hb-α and brain hypoxia was conducted to show their linear correlation. Five mice in each group (*n* = 5) were used for the aging-related studies; six mice in each group (*n* = 6) were used for the Hb-α knockdown studies. The sample size for each experiment was determined by a power analysis (α = 0.05, β = 0.02) of our pilot experiments, and was based on previous work by our group [46]. A value of *p* < 0.05 was considered statistically significant.

## Figures and Tables

**Figure 1 ijms-23-05360-f001:**
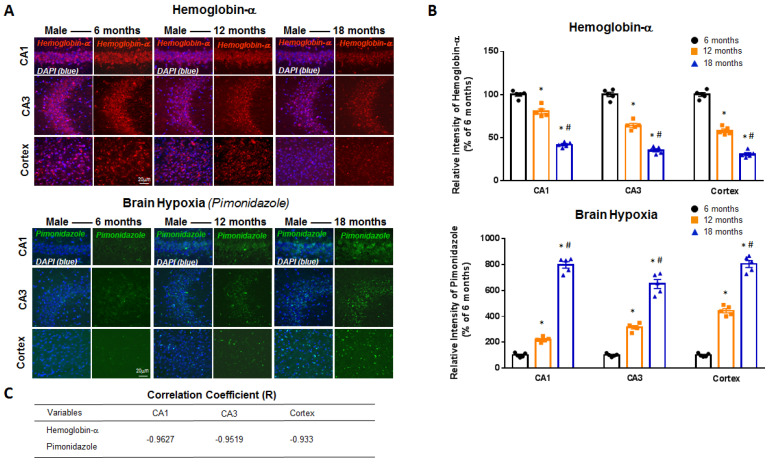
Age-related decline of hemoglobin-α (Hb-α) with parallel increase in brain hypoxia in the intact male mouse brain. (**A**) Representative photomicrographs from male mouse hippocampal CA1, CA3 and cerebral cortex regions showing age-related decline in Hb-α staining and parallel increase in brain hypoxia as measured by pimonidazole (hypoxyprobe-1) staining. (**B**) Semi-quantitative intensity analysis of Hb-α and pimonidazole staining in hippocampal CA1 and CA3 regions and cerebral cortex of all male mice at different ages. (**C**) Correlation coefficient analysis between Hb-α and pimonidazole in hippocampal CA1, CA3 and cortex regions. *: *p* < 0.05 vs. 6 months; #: *p* < 0.05 vs. 12 months; *n* = 5. Scale bar: 20 μm.

**Figure 2 ijms-23-05360-f002:**
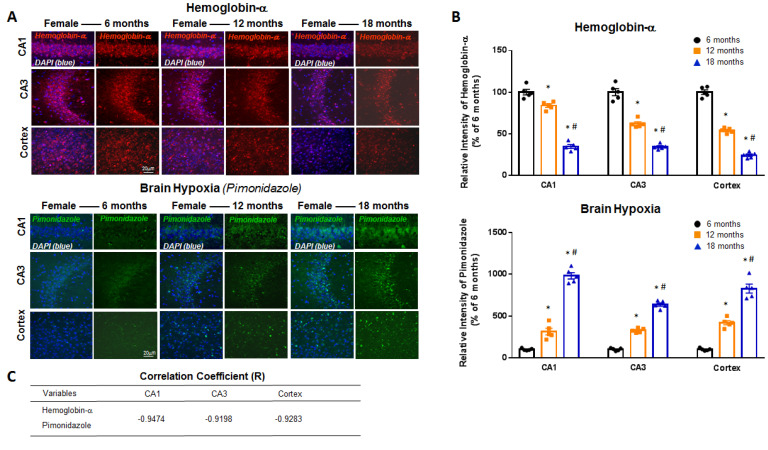
Age-related decline of hemoglobin-α (Hb-α) with parallel increase in brain hypoxia in the intact female mouse brain. (**A**) Representative photomicrographs from female mouse hippocampal CA1, CA3, and cerebral cortex regions showing age-related decline in Hb-α staining and parallel increase in brain hypoxia as measured by pimonidazole (hypoxyprobe-1) staining. (**B**) Semi-quantitative intensity analysis of Hb-α and pimonidazole staining in hippocampal CA1 and CA3 regions and cerebral cortex of all female mice at different ages. (**C**) Correlation coefficient analysis between Hb-α and pimonidazole in hippocampal CA1, CA3 and cortex regions. *: *p* < 0.05 vs. 6 months; #: *p* < 0.05 vs. 12 months; *n* = 5. Scale bar: 20 μm.

**Figure 3 ijms-23-05360-f003:**
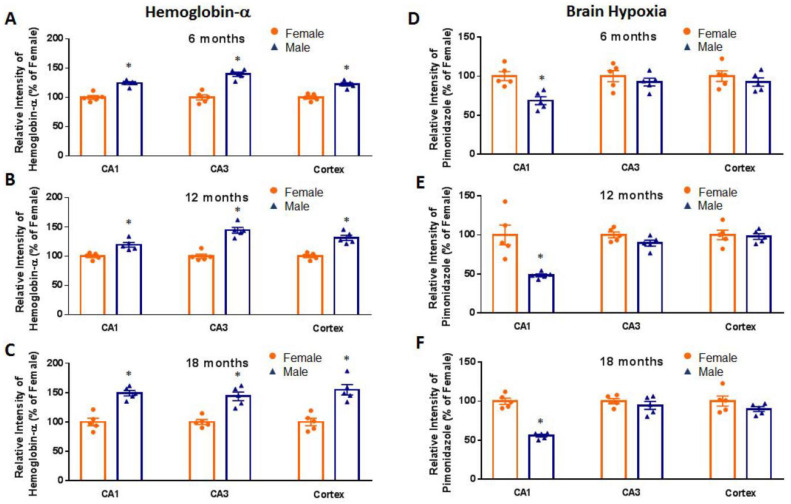
Gender differences in neuronal hemoglobin-α (Hb-α) levels and brain hypoxia in the mouse brain in aging. Semi-quantitative intensity analysis of Hb-α (**A**–**C**) and pimonidazole staining (**D**–**F**) in hippocampal CA1 and CA3 regions and cerebral cortex of all female and male mice at different ages. *: *p* < 0.05 vs. female; *n* = 5.

**Figure 4 ijms-23-05360-f004:**
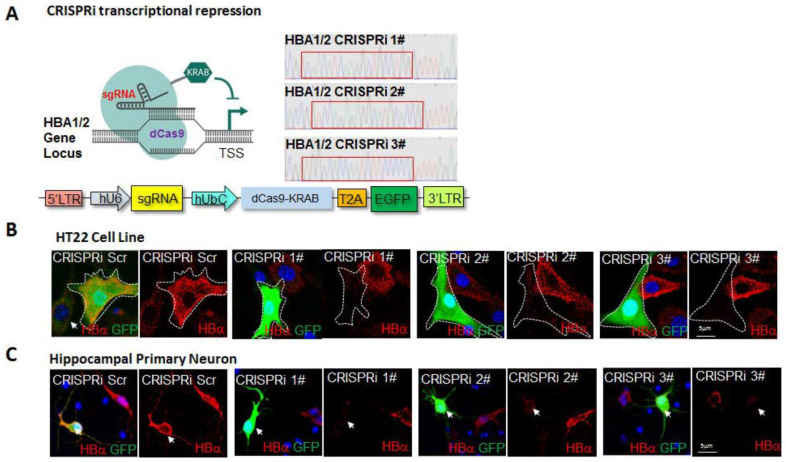
CRISPRi efficiently inactivates HBA1/2 (Hb-α) expression in HT-22 cell line and hippocampal primary neurons. (**A**). Schematic demonstration of the CRISPRi system. (**B**,**C**). Representative immunofluorescence images showing inhibition of Hb-α expression in CRISPRi (#1–3) transfected HT-22 cell line (**B**) and mouse hippocampal primary neurons (**C**). Arrows in (**B**,**C**) indicate non-infected cells in the same well, which strongly express Hb-α. Scale bar: 5 μm.

**Figure 5 ijms-23-05360-f005:**
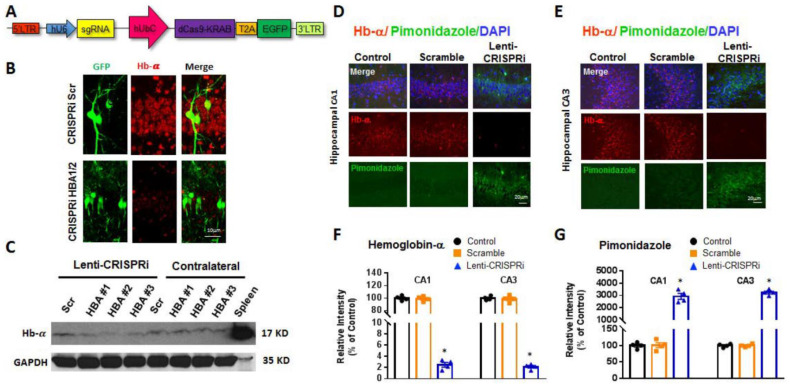
Lenti-CRISPRi efficiently inactivates HBA1/2 (Hb-α) expression in mouse hippocampal CA1 neurons with a resultant increase in hypoxia. (**A**). Schematic demonstration of the conditional CRISPRi system used to selectively suppress HBA1/2 expression in the hippocampal CA1 and CA3 regions. (**B**). Representative IHC for Hb-α in hippocampal CA1 region at 2 weeks after infusion with scramble (upper) or HBA1/2 (lower) lenti-CRISPRi. (**C**). Western Blot analysis showing effectiveness of HBA1/2 lenti-CRISPRi knockdown of Hb-α in mouse hippocampus at 2 weeks after infusion. (**D**–**G**). HBA1/2-lenti-CRISPRi effectively knocked down Hb-α in the hippocampal CA1 and CA3 regions, an effect that correlated with significantly increased CA1 and CA3 hypoxia as measured by pimonidazole staining at 2 weeks after lentivirus infusion (*: *p* < 0.05, *n* = 6). Scale bar: (**B**), 10 μm; (**D**,**E**), 20 μm.

**Figure 6 ijms-23-05360-f006:**
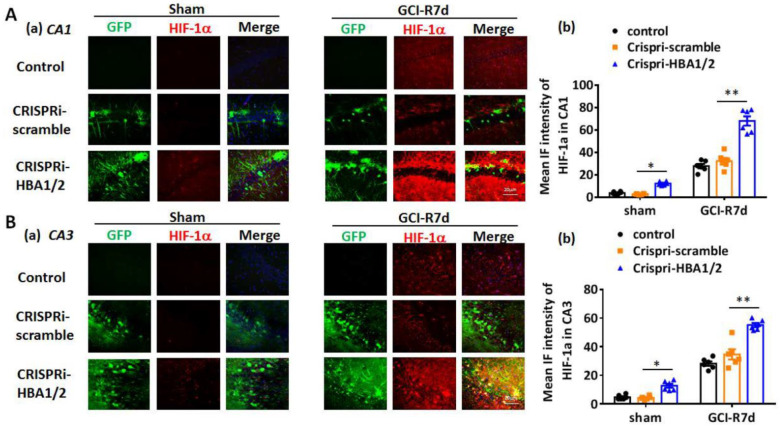
Effect of hemoglobin-α knockdown on HIF-1α expression in the male mouse hippocampal CA1 and CA3 regions following global cerebral ischemia (GCI). (**A**)/(**a**) and (**B**)/(**a**) Representative photomicrographs of HIF-1α immunostaining in 3-month-old male mouse hippocampal CA1 and CA3 regions from control (empty, upper row), scramble (lenti-CRISPRi-scramble, middle row) or Hb-α knockdown (lenti-CRISPRi-HBA1/2, lower row) groups. (**A**)/(**b**) and (**B**)/(**b**) Quantification of mean immunofluorescence (IF) demonstrates a mild increase in HIF-1α expression in sham after Hb-α knockdown (*: *p* < 0.05, *n* = 6). At 7 days after GCI reperfusion (R7d), Hb-α knockdown robustly increased HIF-1α expression in both hippocampal CA1 (**A**) and CA3 regions (**B**). (*: *p* < 0.05, **: *p* < 0.01, *n* = 6). Scale bar: 20 μm.

**Figure 7 ijms-23-05360-f007:**
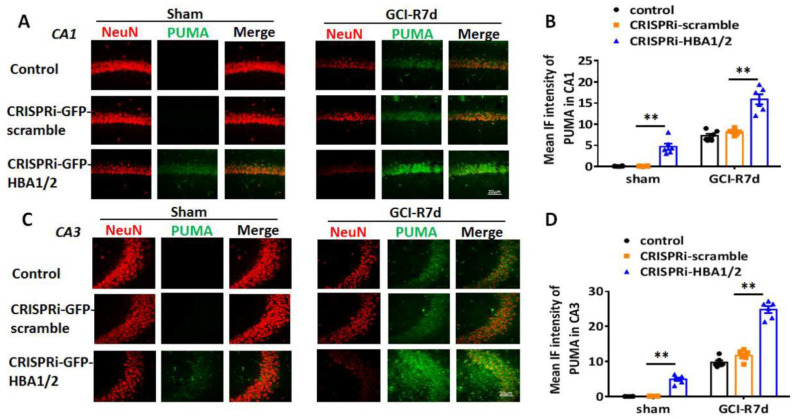
Effect of hemoglobin-α knockdown on expression of the HIF-1α-regulated pro-apoptotic gene, PUMA in the male mouse hippocampal CA1 and CA3 regions following global cerebral ischemia (GCI). (**A**,**C**) Representative photomicrographs of PUMA immunostaining in 3-month-old male mouse hippocampal CA1 (**A**) and CA3 (**C**) regions from control (empty, upper row), scramble (lenti-CRISPRi-scramble, middle row) or Hb-α knockdown (lenti-CRISPRi-HBA1/2, lower row) groups. Staining in red represents NeuN, a neuronal marker, and staining in green represents PUMA. (**B**,**D**) Quantification of mean immunofluorescence (IF) demonstrates a mild increase in PUMA intensity in sham after Hb-α knockdown. At 7 days after GCI reperfusion (R7d), Hb-α knockdown robustly increased PUMA intensity in the hippocampal CA1 and CA3 regions (**: *p* < 0.01, *n* = 6). Scale bar: 20 μm.

**Figure 8 ijms-23-05360-f008:**
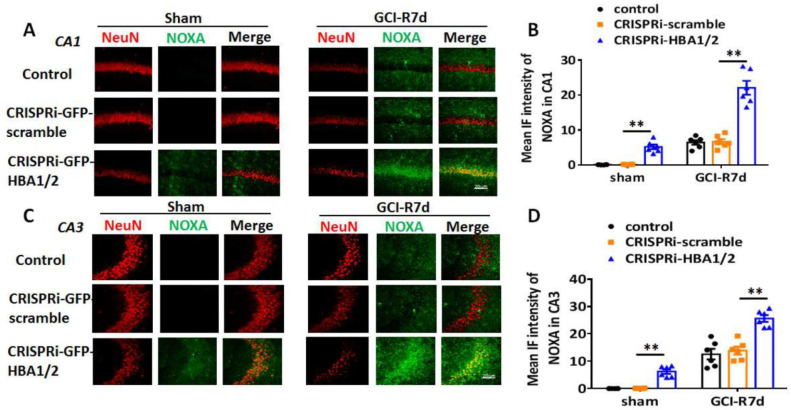
Effect of hemoglobin-α knockdown on expression of the HIF-1α-regulated pro-apoptotic gene, NOXA in the male mouse hippocampal CA1 and CA3 regions following global cerebral ischemia (GCI). (**A**,**C**) Representative photomicrographs of NOXA immunostaining in 3-month-old male mouse hippocampal CA1 and CA3 regions from control (empty, upper row), scramble (lenti-CRISPRi-scramble, middle row) or Hb-α knockdown (lenti-CRISPRi-HBA1/2, lower row) groups. (**B**,**D**). Quantification of mean immuno-fluorescence (IF) demonstrates a mild increase in NOXA expression in sham after Hb-α knockdown (**: *p* < 0.01, *n* = 6). At 7 days after GCI reperfusion (R7d), Hb-α knockdown robustly increased NOXA expression in both CA1 (**A**) and CA3 regions (**C**). (**: *p* < 0.01, *n* = 6). Scale bar: 20 μm.

**Figure 9 ijms-23-05360-f009:**
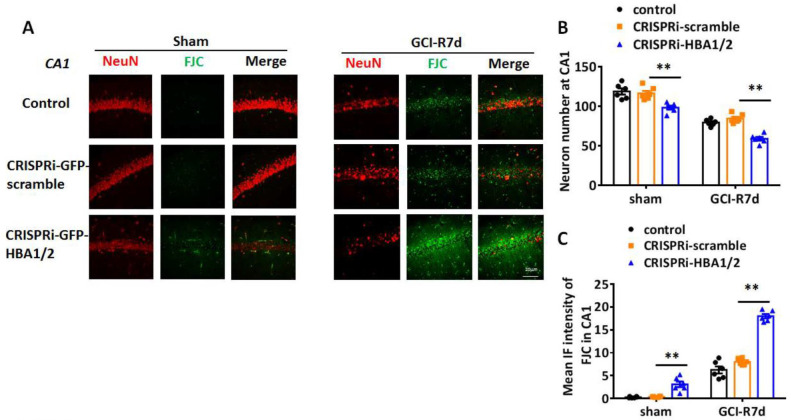
Effect of hemoglobin-α knockdown on neurodegeneration in the male mouse hippocampal CA1 region following global cerebral ischemia (GCI). (**A**) Representative photomicrographs from male mouse hippocampal CA1 region showing staining with the neuronal marker NeuN or with the neurodegeneration marker Fluoro Jade C (FJC). (**B**) NeuN-positive cell number in hippocampal CA1 region of various groups. (**C**) Intensity analysis of FJC staining in hippocampal CA1 region of various groups. **: *p* < 0.01. Scale bar: 20 μm.

**Figure 10 ijms-23-05360-f010:**
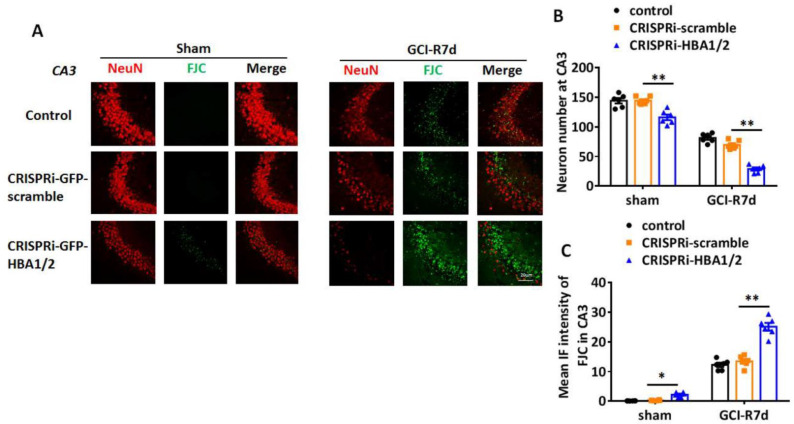
Effect of hemoglobin-α knockdown on neurodegeneration in the male mouse hippocampal CA3 region following global cerebral ischemia (GCI). (**A**) Representative photomicrographs from male mouse hippocampal CA3 region showing staining with the neuronal marker NeuN or with the neurodegeneration marker Fluoro Jade C (FJC). (**B**) NeuN-positive cell number in hippocampal CA3 region of various groups. (**C**) Intensity analysis of FJC staining in hippocampal CA3 region of various groups. *: *p* < 0.05, **: *p* < 0.01. Scale bar: 20 μm.

## Data Availability

No new data sets were created or analyzed in this study. Data sharing is not applicable to this article.

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
