# Peer review of "Regulation and Role of Neuron-Derived Hemoglobin in the Mouse Hippocampus"

_ijms, 2022, doi:10.3390/ijms23105360_

Round 1
Reviewer 1 Report
Summary
In the manuscript titled ‘Regulation and Role of Neuron-Derived Hemoglobin in the Mouse Hippocampus’ the authors have determined the expression of hemoglobin-alpha in different parts of the hippocampus in mice. They also investigate the role of hemoglobin-alpha in hypoxia, neuronal survival, and neuronal degeneration. This was achieved by knocking down the expression of hemoglobin-alpha in the hippocampus using CRISPR-Cas9 techniques. Most findings were based on immunohistochemical studies.
Strengths
The authors have explained the basis and the rationale of the study well. They have appropriate controls in their experiments and have a robust model. The manuscript is written well and concepts are clearly explained.
Weaknesses
The authors can make the manuscript stronger by addressing the following.
- It is not clear why the authors chose the CA1 and CA3 regions of the hippocampus. Please provide a rationale to the reader.
- In many places, there is a quantification of immunohistochemistry images. It is not clear how the images were quantified. Please describe the method used to perform this quantification adequately.
- In page 5, line 213, the authors mention about correlations between hemoglobin-alpha levels and brain hypoxia in different regions. This statement will be further strengthened by performing a correlation analysis that would include the calculation of a correlation coefficient. This can easily be done with the data already present.
- In figure 4B and 4C, the authors should show individual fluorescent channels and then a merged one if they want. By showing only the merged channels, it is difficult to determine the level of hemoglobin-alpha as in some cases the expression of GFP is very bright. This interferes with the ability to determine the expression of hemoglobin-alpha.
- In the experiments performed that are related to figure 5, the age of the mice is not clear. Please specify the age of the mice.
- In figure 5D and E, please label the figure in such a way that each row is labeled with the protein that is being detected. The current way the figure is labeled requires the reader to guess based on colors. This will be difficult for readers who may be color blind.
- In figure 7 onwards, the bar graphs are depicted to show the distribution of the individual mice. It would be better to show all bar graphs in this manner.
Some minor changes recommended are:
- It is unclear why one of the affiliations is in bold. Please remove the bold feature.
- There is a change in the font and color in various places of the manuscript. Please make the color and font consistent throughout the manuscript.
- In the materials and methods section, the Lentivirus production sub-section has the centrifugation speeds in rpm in some places and ‘g’ in other places. Please report the speed in rcf or ‘g’ so that it is easier for other researchers to reproduce it.
By incorporating these changes, it will be easier for the reader to understand the work outlined in the manuscript.
Author Response
We thank the editors and reviewers for the insightful comments and constructive suggestions. We have revised the relevant segments of the original manuscript. Below, we provide our point-by-point responses to the reviewers’ concerns and indicate relevant locations of incorporation of changes or additions in the manuscript. In the article file, all changes are indicated in red font.
Reviewer 1
The authors can make the manuscript stronger by addressing the following.
Comment 1. It is not clear why the authors chose the CA1 and CA3 regions of the hippocampus. Please provide a rationale to the reader.
Response: The hippocampal CA1 and CA3 in the brain and are highly susceptible to ischemic damage under hypoperfusion. We have now stated the rationale in the Introduction. (PMID 3835581. Page 3-4. Line 66-69)
Comment 2. In many places, there is a quantification of immunohistochemistry images. It is not clear how the images were quantified. Please describe the method used to perform this quantification adequately.
Response: Thanks. The method of quantification of immunohistochemistry images is stated in the methods (page 8. Line 171-175)
Comment 3. In page 5, line 213, the authors mention about correlations between hemoglobin-alpha levels and brain hypoxia in different regions. This statement will be further strengthened by performing a correlation analysis that would include the calculation of a correlation coefficient. This can easily be done with the data already present.
Response: Thanks. We have performed the requested correlation coefficient analysis on Figures 1 and 2 and it showed a profound negative correlation between neuronal Hb-α levels and brain hypoxia. We have now revised Figures 1 and 2 to add the results of the correlation coefficient analysis (see Figures 1&2 C). In addition, we added description of the analysis to the “Methods” section (see page 9, lines. 197-198), and description of the analysis results to the “Results” section (see page10, lines 226-229).
Comment 4. In figure 4B and 4C, the authors should show individual fluorescent channels and then a merged one if they want. By showing only the merged channels, it is difficult to determine the level of hemoglobin-alpha as in some cases the expression of GFP is very bright. This interferes with the ability to determine the expression of hemoglobin-alpha.
Response: As suggested by the reviewer, we have added the single channel of hemoglobin-a (red) to Figures 4B and 4C.
Comment 5. In the experiments performed that are related to figure 5, the age of the mice is not clear. Please specify the age of the mice.
Response: Thanks. The specific age of the mice is stated in the legend of figure 5 and methods (Page 4, line 86-89)
Comment 6. In figure 5D and E, please label the figure in such a way that each row is labeled with the protein that is being detected. The current way the figure is labeled requires the reader to guess based on colors. This will be difficult for readers who may be color blind.
Response: The labels for figure 5D and E have been adjusted, different colors have been indicated.
Comment 7: In figure 7 onwards, the bar graphs are depicted to show the distribution of the individual mice. It would be better to show all bar graphs in this manner.
Response: Thanks. The data presentation throughout the figures are now consistent. We show “bar and scatted dot”, and each dot represents data of individual mice.
Some minor changes recommended are:
- It is unclear why one of the affiliations is in bold. Please remove the bold feature.
- There is a change in the font and color in various places of the manuscript. Please make the color and font consistent throughout the manuscript.
Response: Thanks, we have adjusted the font and colors, and made them consistent.
In the materials and methods section, the Lentivirus production sub-section has the centrifugation speeds in rpm in some places and ‘g’ in other places. Please report the speed in rcf or ‘g’ so that it is easier for other researchers to reproduce it.
Response: Because a swing bucket is used for the ultracentrifuge, the g force for one sample is within a range according to the radius. We have added the range of “g” information in the methods (Page 5. Line 109, 112)

Reviewer 2 Report
This study investigates the role of neuron derived hemoglobin in hippocampus which could be of great benefit to the medical community. Some minor edits can be seen below:
Introduction:
-It is mentioned that a deficiency of hemoglobin is seen in Parkinson's, Alzheimer's, and dementia. Is there a way based on the level of deficiencies to differentiate between the various diseases or have studies been conducted on this?
Methods:
-How many mice were utilized for this study? I would like to know that the findings have a large enough sample size to make them statistically significant.
Author Response
We thank the editors and reviewers for the insightful comments and constructive suggestions. We have revised the relevant segments of the original manuscript. Below, we provide our point-by-point responses to the reviewers’ concerns and indicate relevant locations of incorporation of changes or additions in the manuscript. In the article file, all changes are indicated in red font.
Reviewer 2
Introduction:
-It is mentioned that a deficiency of hemoglobin is seen in Parkinson's, Alzheimer's, and dementia. Is there a way based on the level of deficiencies to differentiate between the various diseases or have studies been conducted on this?
Response. To our knowledge, this specific question has not been examined to date.
Methods:
-How many mice were utilized for this study? I would like to know that the findings have a large enough sample size to make them statistically significant.
Response. The number of animals used in each experiment was demonstrated as “N” number in the figure legends. We have now also stated the animal number in the Methods under the “Statistical Analysis” section, with a rationale to explain how we determined the animal number to be used. (Page 10, Line 198-201)
